# Mitochondrial DNA and Inflammation Are Associated with Cerebral Vessel Remodeling and Early Diabetic Kidney Disease in Patients with Type 2 Diabetes Mellitus

**DOI:** 10.3390/biom14040499

**Published:** 2024-04-19

**Authors:** Ligia Petrica, Florica Gadalean, Danina Mirela Muntean, Dragos Catalin Jianu, Daliborca Vlad, Victor Dumitrascu, Flaviu Bob, Oana Milas, Anca Suteanu-Simulescu, Mihaela Glavan, Sorin Ursoniu, Lavinia Balint, Maria Mogos-Stefan, Silvia Ienciu, Octavian Marius Cretu, Roxana Popescu, Cristina Gluhovschi, Lavinia Iancu, Adrian Vlad

**Affiliations:** 1Department of Internal Medicine II, Division of Nephrology, “Victor Babes” University of Medicine and Pharmacy, No. 2, Eftimie Murgu Sq., 300041 Timisoara, Romania; petrica.ligia@umft.ro (L.P.); bob.flaviu@umft.ro (F.B.); milas.oana@umft.ro (O.M.); anca.simulescu@umft.ro (A.S.-S.); patruica.mihaela@umft.ro (M.G.); lavinia.balint@umft.ro (L.B.); maria.mogos@umft.ro (M.M.-S.); silvia-ioana.ienciu@umft.ro (S.I.); gluh@umft.ro (C.G.); iuliana.alioani@umft.ro (L.I.); 2Centre for Molecular Research in Nephrology and Vascular Disease, Faculty of Medicine, “Victor Babes” University of Medicine and Pharmacy, No. 2, Eftimie Murgu Sq., 300041 Timisoara, Romania; daninamuntean@umft.ro (D.M.M.); jianu.dragos@umft.ro (D.C.J.); vlad.daliborca@umft.ro (D.V.); dumitrascu.victor@umft.ro (V.D.); sursoniu@umft.ro (S.U.); popescu.roxana@umft.ro (R.P.); vlad.adrian@umft.ro (A.V.); 3Centre for Cognitive Research in Neuropsychiatric Pathology (Neuropsy-Cog), Faculty of Medicine, “Victor Babes” University of Medicine and Pharmacy, No. 2, Eftimie Murgu Sq., 300041 Timisoara, Romania; 4Center for Translational Research and Systems Medicine, Faculty of Medicine, “Victor Babes” University of Medicine and Pharmacy, No. 2, Eftimie Murgu Sq., 300041 Timisoara, Romania; 5County Emergency Hospital Timisoara, 300723 Timisoara, Romania; 6Department of Functional Sciences III, Division of Pathophysiology, “Victor Babes” University of Medicine and Pharmacy, No. 2, Eftimie Murgu Sq., 300041 Timisoara, Romania; 7Department of Neurosciences VIII, Division of Neurology I, “Victor Babes” University of Medicine and Pharmacy, No. 2, Eftimie Murgu Sq., 300041 Timisoara, Romania; 8Department of Biochemistry and Pharmacology IV, Division of Pharmacology, “Victor Babes” University of Medicine and Pharmacy, No. 2, Eftimie Murgu Sq., 300041 Timisoara, Romania; 9Department of Functional Sciences III, Division of Public Health and History of Medicine, “Victor Babes” University of Medicine and Pharmacy, No. 2, Eftimie Murgu Sq., 300041 Timisoara, Romania; 10Department of Surgery I, Division of Surgical Semiology I, “Victor Babes” University of Medicine and Pharmacy, No. 2, Eftimie Murgu Sq., 300041 Timisoara, Romania; cretu.marius@umft.ro; 11Emergency Clinical Municipal Hospital Timisoara, 300041 Timisoara, Romania; 12Department of Microscopic Morphology II, Division of Cell and Molecular Biology II, “Victor Babes”, University of Medicine and Pharmacy, No. 2, Eftimie Murgu Sq., 300041 Timisoara, Romania; 13Department of Internal Medicine II, Division of Diabetes, Nutrition, and Metabolic Diseases, “Victor Babes” University of Medicine and Pharmacy, No. 2, Eftimie Murgu Sq., 300041 Timisoara, Romania

**Keywords:** mitochondrial DNA, inflammation, cerebral vessels, diabetic kidney disease

## Abstract

Cerebrovascular disease accounts for major neurologic disabilities in patients with type 2 diabetes mellitus (DM). A potential association of mitochondrial DNA (*mtDNA*) and inflammation with cerebral vessel remodeling in patients with type 2 DM was evaluated. A cohort of 150 patients and 30 healthy controls were assessed concerning urinary albumin/creatinine ratio (UACR), synaptopodin, podocalyxin, kidney injury molecule-1 (KIM-1), N-acetyl-β-(D)-glucosaminidase (NAG), interleukins IL-17A, IL-18, IL-10, tumor necrosis factor-alpha (TNFα), intercellular adhesion molecule-1 (ICAM-1). *MtDNA-CN* and nuclear DNA (*nDNA*) were quantified in peripheral blood and urine by qRT-PCR. Cytochrome b (*CYTB*) gene, subunit 2 of NADH dehydrogenase (*ND2*), and beta 2 microglobulin nuclear gene (*B2M*) were assessed by TaqMan assays. *mtDNA-CN* was defined as the ratio of the number of *mtDNA/nDNA* copies, through analysis of the *CYTB/B2M* and *ND2/B2M* ratio; cerebral Doppler ultrasound: intima-media thickness (IMT)—the common carotid arteries (CCAs), the pulsatility index (PI) and resistivity index (RI)- the internal carotid arteries (ICAs) and middle cerebral arteries (MCAs), the breath-holding index (BHI). The results showed direct correlations of CCAs-IMT, PI-ICAs, PI-MCAs, RI-ICAs, RI-MCAs with urinary *mtDNA*, IL-17A, IL-18, TNFα, ICAM-1, UACR, synaptopodin, podocalyxin, KIM-1, NAG, and indirect correlations with serum *mtDNA*, IL-10. BHI correlated directly with serum IL-10, and serum *mtDNA*, and negatively with serum IL-17A, serum ICAM-1, and NAG. In neurologically asymptomatic patients with type 2 DM cerebrovascular remodeling and impaired cerebrovascular reactivity may be associated with *mtDNA* variations and inflammation from the early stages of diabetic kidney disease.

## 1. Introduction

Diabetic kidney disease (DKD) is recognized as the leading cause of chronic kidney disease and may be attributed to approximately 40% of patients with type 2 diabetes mellitus (DM) and 30% of patients with type 1 DM being referred to renal replacement therapies worldwide [1].

Among the multiple macro- and microvascular complications of DM, cerebrovascular disease accounts for major motor and cognitive disabilities [2] and is associated with high post-stroke mortality [3]. The increased burden imposed by cerebrovascular disease in type 2 DM relies upon cerebral vessels’ complex vascular remodeling which consists of atherosclerosis, arteriosclerosis, and cerebral microangiopathy [4]. Subclinical atherosclerosis is highly prevalent in patients with DKD as compared to non-diabetic subjects, independently of traditional cardiovascular risk factors [5]. It has been reported that asymptomatic carotid artery stenosis ≥50% imposes a very high risk for major adverse vascular events in patients with type 2 DM [6].

The intimate pathways that coordinate the brain-kidney axis in the course of type 2 DM rely upon endothelial dysfunction, oxidative stress, inflammation, and mitochondrial dysfunction, which may reflect a generalized vascular involvement [7,8].

Mitochondrial dysfunction is an important pathogenic mechanism implicated in the development and progression of DKD [9] through complex alterations in mitochondrial functions, including mitochondrial DNA (mtDNA) mutations and mtDNA copy numbers depletion [10,11]. Damaged mitochondria deriving from tubular and glomerular cells release their content into the extracellular space and subsequently into the systemic circulation. Hence, mtDNA fragments deriving from the systemic circulation are filtered through the glomeruli and actively secreted into the urine. Extracellular mtDNA may be detected and quantified in both serum and urine and may be utilized as a biomarker of mitochondrial dysfunction [12].

Mitochondrial DNA changes in serum and urine display a specific signature in relation to inflammation within the diabetic kidney at the glomerular and tubular levels, even in patients with normoalbuminuric type 2 DM [13]. Mitochondrial dysfunction, due to mtDNA damage and subsequent excessive production of reactive oxygen species, leads to the initiation and progression of atherosclerosis [14,15,16] and may promote an increased risk for ischemic stroke [17]. Mitochondrial DNA alterations and their interconnection with inflammation contribute to extensive atherosclerotic processes [18,19] Chronic inflammatory mechanisms have a key role in the asymptomatic stages of atherosclerosis and involve the expression of adhesion molecules and proinflammatory cytokines [20,21,22,23].

Cerebral vessel remodeling within the confines of DM may be assessed by neurosonologic methods that allow for a detailed analysis of ultrasonographic parameters, such as carotid intima-media thickness (IMT), which is a marker of subclinical atherosclerosis. The increased values of IMT are considered a surrogate marker for cardiovascular and cerebrovascular risk [24,25], even in normoalbuminuric patients with type 2 DM [26,27]. The pulsatility index (PI) and the resistivity index (RI) are hemodynamic indices that evaluate increased resistance in the examined vessels due to cerebral vessel remodeling. The examined arteries, such as the internal carotid arteries (ICAs) and the middle cerebral arteries (MCAs), which are low-resistance vessels, display increased PIs and RIs either as a consequence of increased resistance or an impaired vasodilation capacity in relation to processes of atherosclerosis, arteriosclerosis, and cerebral microangiopathy [25,27,28].

Cerebrovascular reactivity (CVR) is a hemodynamic parameter that represents the increase in normal blood flow velocity to a vasodilatory stimulus [29]. In patients with type 2 DM, the CVR is impaired due to preexisting vasodilation, a fact that translates into a reduced capacity of the mechanisms of hemodynamic autoregulation at variations in the cerebral blood flow [27,30,31]. 

Despite a large body of evidence with regard to the complex relationship between mtDNA abnormalities and atherosclerosis, the literature lacks data concerning the intervention of mtDNA in cerebral vessel structural modifications in type 2 DM. To date, no clinical or experimental study has approached cerebrovascular remodeling in relation to mtDNA variations in the absence of a history or symptoms of cerebrovascular disease.

To the best of our knowledge, this is the first study that aims to evaluate, from a molecular perspective, a potential association of mtDNA changes in blood and urine and biomarkers of inflammation with cerebral vessel structural and functional modifications in neurologically asymptomatic patients with type 2 DM.

## 2. Materials and Methods

### 2.1. Cohort/Inclusion/Exclusion Criteria

A cohort of consecutive 220 patients with type 2 DM attending the Outpatient Department of Nephrology and the Outpatient Department of Diabetes and Metabolic Diseases (from January 2021 through December 2022), ages 50–78 years, were screened for the study according to personal visits and chart reviews. The inclusion criteria were DM duration over 5 years and therapy with oral antidiabetic medication (metformin, gliclazide) or insulin, angiotensin-converting enzyme inhibitors or angiotensin receptor blockers, and statins. The exclusion criteria were poor glycemic control (HbA_1c_ > 10%) and a history or symptoms of cerebrovascular disease. Out of the 220 patients screened, a cohort of 150 patients [52 patients with normoalbuminuria (urinary albumin: creatinine ratio-UACR < 30 mg/g), 48 patients with microalbuminuria (UACR 30–300 mg/g), and 50 patients with macroalbuminuria (UACR > 300 mg/g)] and 30 age- and gender-matched healthy controls enrolled in the records of a general practitioner with no history of renal diseases and for whom DM or pre-diabetes were excluded by a level of HbA_1c_ ≤ 5.6% (group 4) were enrolled in this case series study (Figure 1).

### 2.2. Assessment of mtDNA

*MtDNA-CN* and nuclear DNA (*nDNA*) were quantified in peripheral blood and urine by qRT-PCR (CFX Connect-Biorad Laboratories, Carlsbad, CA, USA). The cytochrome b (*CYTB*) gene, subunit 2 of NADH dehydrogenase (*ND2*), and beta 2 microglobulin nuclear gene (*B2M*) were assessed by TaqMan assays. Primers for the *CYTB* and *ND2* genes were utilized as target sequences for the assessment of *mtDNA*. *B2M* was utilized as an internal reference gene for *nDNA* analysis. *MtDNA-CN* was defined as the ratio of the number of *mtDNA/nDNA* copies through analysis of the *CYTB/B2M* and *ND2/B2M* ratios. Genomic DNA was obtained from biological samples using PureLink™ Genomic DNA Kit (Life Technologies, Carlsbad, CA, USA), following the manufacturer’s instructions. The concentration of extracted DNA was measured by fluorimetric quantification (Qubit, Invitrogen, Thermo Fisher Scientific, Waltham, MA, USA). For real-time quantitative tests, the DNA samples were diluted to 10 ng/µL. Real-time quantitative polymerase chain reaction was applied using TaqMan Universal PCR master mix and TaqMan primers (Applied Biosystems, Thermo Fisher Scientific, Waltham, MA, USA). Samples were run in triplicate, in MicroAmp^®^ Optical 96-well reaction plates, each well containing 9 µL of diluted DNA, 1 µL primers, and 10 µL of master mix. The total reaction volume was 20 μL. The thermal cycle profile was 2 min at 500C for UNG incubation, 10 min at 950C for polymerase activation, and 40 cycles of 15 s at 950C (denaturation) and 1 min at 600C (annealing and extension). For each run, a melting curve analysis to check nonspecific products. 

Relative *mtDNA* quantification was carried out as previously described [32]. The results were analyzed using the comparative Ct method. ΔCt (values of Δ cycle thresholds) in the sample were calculated by subtracting the values for the reference gene from the sample Ct and normalizing to nuclear DNA. 2-ΔCt was obtained, and the results were expressed as relative quantification. The obtained values were normalized to nuclear DNA and are reported as the number of copies per nuclear DNA (*mtDNA/nDNA*). Urinary values were normalized to urine creatinine. 

### 2.3. Laboratory Assessments

The serum and urinary specimens of patients and controls were frozen at −80 °C and thawed before assay. Urinary biomarkers were evaluated in the same first-morning urine sample and reported as per the urinary creatinine ratio. The biomarkers studied were assessed by the ELISA technique, as follows: podocyte injury biomarkers synaptopodin (Catalogue Nr. abx055120 Abbexa, Cambridge, UK; sensitivity—0.10ng/mL, detection range—0.156–10 ng/mL, coefficient of variance (CV) < 10%) and podocalyxin (Catalogue Nr. E-EL-H2360 Elabscience, Houston, TX, USA; sensitivity—0.1 ng/mL, detection range—0.16–10 ng/mL, coefficient of variance (CV) < 10%); proximal tubule (PT) dysfunction biomarkers kidney injury molecule—1(KIM-1, Catalogue Nr. E-EL-H6029 Elabscience, Houston, TX, USA; sensitivity—4.69 pg/mL, detection range—7.81–500 pg/mL, CV < 10%) and N-acetyl-β-(D)-glucosaminidase (NAG, Catalogue Nr. E-EL-H0898 Elabscience, Houston, TX, USA; sensitivity—0.94 ng/mL, detection range—1.56–100 ng/mL, CV < 10%); biomarkers of inflammation: tumor necrosis alpha (TNFα, Catalogue Nr. E-EL-H0109 Elabscience, Houston, TX, USA; sensitivity—4.69 pg/mL, detection range—7.81–500 pg/mL, CV < 10%); intercellular adhesion molecule (ICAM-1, Catalogue Nr. E-EL-H6114 Elabscience, Houston, TX, USA; sensitivity—0.19 ng/mL, detection range—0.31–20 ng/mL, CV < 10%); interleukins [IL-17A (Catalogue Nr. E-EL-H0105 Elabscience, Houston, TX, USA; sensitivity—18.75 pg/mL, detection range—31.25–2000 pg/mL, CV < 10%), IL-18 (Catalogue Nr. E-EL-H0253 Elabscience, Houston, TX, USA; sensitivity—9.38 pg/mL, detection range—15.63–1000 pg/mL, CV < 10%), and IL-10 (Catalogue Nr. E-EL-H6154 Elabscience, Houston, TX, USA; sensitivity—0.94 pg/mL, detection range—1.56–100 pg/mL, CV < 10%)]. All serum and urinary samples were run in triplicate and assessed according to the instructions in the manufacturer’s brochure. Further, eGFR was calculated with the combined serum creatinine-cystatin C (CKD-EPI equation), according to the KDIGO 2024 Guideline for the Evaluation and Management of Chronic Kidney Disease [33].

### 2.4. Neurosonologic Ultrasound Assessment

All cerebrovascular hemodynamic indices were evaluated by an experienced neurologist blinded for the clinical and biological data of the patients and healthy controls by an ultrasound machine with high resolution (Esaote MyLab 8, Genoa, Italy), equipped with a Color Ultrasound System, which included two transducers: one with a selectable minimum frequency band between 1.7 MHz and 4 MHz (multifrequency sectorial transducer-phased array) and another probe with a frequency from 3.6 MHz to 16 MHz (linear transducer). The cerebrovascular ultrasound technique was applied as presented previously [34]. Briefly, we will describe the methods utilized. 

#### 2.4.1. Carotid Artery Intima-Media Thickness (IMT)

Carotid artery IMT was evaluated bilaterally in the common carotid arteries (CCAs). The common carotid artery IMT represents the distance between the luminal-intimal interface and the media-adventitial interface of the carotid arteries. This is a double-line pattern displayed by the ultrasound method in brightness mode (B-mode) on the wall of the carotid artery in a longitudinal phase [24]. All patients and control subjects were assessed by three IMT measurements, and the median value was utilized for further analysis. The normal cut-off point was appreciated at <1.0 mm. 

#### 2.4.2. Pulsatility Index (PI) and Resistivity Index (RI)

The pulsatility indices (PI) and the resistivity indices (RI) were evaluated bilaterally in the ICAs by means of a continuous wave of 4 MHz-CW, by extracranial Doppler ultrasound, and in the MCAs with a pulsed wave of 2 MHz-PW, by transcranial Doppler ultrasound. These hemodynamic indices were calculated automatically by specific formulas, such as the following: Gosling’s PI = (systolic flow velocity − diastolic flow velocity)/mean flow velocity (normal value < 1); Pourcellot’s RI = (systolic flow velocity − diastolic flow velocity/systolic flow velocity) (normal value < 0.7) [25].

#### 2.4.3. Cerebrovascular Reactivity

The CVR, which is the vasomotor reactivity of cerebral vessels in response to a vasodilation stimulus, was assessed by a breath-holding maneuver, namely the transcranial Doppler breath-holding test (BHT) in the MCAs, bilaterally. The procedure is initiated after normal breathing of room air for approximately 4 min, followed by holding breath for 30 s by the end of a normal inspiration. The hemodynamic parameters (mean flow velocity—MFV, systolic flow velocity, and diastolic flow velocity) were monitored at rest, during the breath-holding maneuver, and by the end of the BHT, at the peak of hypercapnia, which was utilized as the vasodilatory stimulus. The maneuver was repeated after 2–3 min of rest in order to allow the MFV to return to basal levels, and the mean MFVs in the MCAs and the mean breath-holding index (BHI) were calculated. 

The BHI is calculated as the percent increase in MFV in the MCAs evaluated by the breath-holding maneuver divided by seconds of breath-holding [(Vbh − Vr/Vr) × 100 s^−1^], where Vbh is the MFV in the MCAs at the end of breath-holding, Vr is the MFV at rest, and s^−1^ represents per second of breath-holding. The normal value for the BHI was set at 1.2 ± 0.6 [23]. 

### 2.5. Statistical Analysis 

Clinical and biological data are presented as medians and interquartile ranges (IQR) for variables with a skewed distribution. The differences between subgroups were analyzed using the Mann–Whitney U test to compare two groups and the Kruskal–Wallis test to compare four groups according to the values’ distribution. This statistical method was applied according to the requirements of case-series studies. Analyses have been conducted to investigate potential associations between cerebrovascular hemodynamic indices and mtDNA in serum and urine, markers of inflammation (interleukins, TNFα, ICAM-1), and podocyte and PT markers.

In addition to the earlier results obtained in patients with DM type 2 and normoalbuminuric DKD [13], other analyses have been carried out to better characterize the possible association of the inflammatory markers (TNFα, ICAM-1) with serum and urinary mtDNA, as well as with biomarkers of podocyte damage and PT dysfunction. 

Regression analyses were performed to assess the significance of the relationship between the cerebrovascular hemodynamic indices and serum and urinary mtDNA, serum and urinary inflammatory markers, and other continuous variables such as synaptopodin, podocalyxin, KIM-1, NAG, UACR, and eGFR. A univariable regression analysis was carried out to evaluate the significance of the relationships between continuous variables in all four groups together (pooled data of normo-, micro-, and macroalbuminuric patients, and healthy controls). Only significant variables provided by univariable regression analysis have been introduced into multivariable regression analysis models. A parsimonious model (few variables) is usually preferred over a complex model (many variables). One way to obtain a parsimonious model from a model with many independent variables consists of iterative removal of the independent variable least significantly related to the dependent variable until all of them are significantly associated with the response variable.

The statistical significance was set at *p* < 0.05 and the analysis was carried out in Stata 18 (StataCorp, College Station, TX, USA).

## 3. Results

### 3.1. Demographic, Clinical, and Biological Data of Patients with Type 2 DM and Healthy Controls

Demographic, clinical, and biological data of patients with type 2 DM and healthy controls are presented in Table 1 as medians and IQR. In Table 1, p values were corrected for multiple comparisons. The data shows significant differences between the groups of patients and controls studied and has been analyzed in part in a study performed by us previously [13]. In addition to the results provided in that study, we introduced supplemental data with regard to two proinflammatory markers with known impact at both renal and vascular levels, namely TNFα and ICAM-1. These markers were increased in serum and urine across all groups of patients vs. healthy control subjects. Furthermore, Table 1 presents the cerebrovascular hemodynamic indices, which show increased values of IMT-CCAs, PI-ICAs, RI-ICAs, PI-MCAs, and RI-MCAs and decreased values of BHI in patients vs. controls.

### 3.2. TNFα and ICAM-1 Changes in Blood and Urine are Associated with Variations of mtDNA and with Biomarkers of Podocyte Injury and PT Dysfunction in Early DKD of Patients with Type 2 DM 

In univariable regression analysis, serum TNFα correlated negatively with eGFR and serum mtDNA (*p* < 0.001) and directly with UACR, synaptopodin, podocalyxin, KIM-1, and NAG (*p* < 0.001). Urinary TNFα correlated directly with urinary mtDNA, UACR, synaptopodin, podocalyxin, KIM-1, and NAG (*p* < 0.001) and indirectly with eGFR (*p* < 0.001). 

Serum ICAM-1 correlated indirectly with serum mtDNA and eGFR (*p* < 0.001) and directly with UACR, synaptopodin, podocalyxin, KIM-1, and NAG (*p* < 0.001). Urinary ICAM-1 correlated directly with urinary mtDNA (*p* < 0.001) and followed the same correlations as serum ICAM-1 with UACR, the biomarkers of podocyte damage and PT dysfunction (*p* < 0.001). Also, urinary ICAM-1 correlated negatively with eGFR (*p* < 0.001).

TNFα and ICAM-1 in blood and urine were significantly increased in patients with type 2 DM, with a progressively ascending trend from normo- to micro- and macroalbuminuria. This trend was associated with decreased levels of serum mtDNA and increased levels of urinary mtDNA.

Multivariable regression analysis yielded models in which serum TNFα correlated directly with UACR, KIM-1, and NAG and negatively with serum mtDNA (R^2^ = 0.893; *p* < 0.0001). Urinary TNFα correlated directly with UACR, KIM-1, NAG, and urinary mtDNA (R^2^ = 0.683; *p* < 0.0001). Serum ICAM-1 displayed a direct correlation with UACR, KIM-1, NAG, synaptopodin, and podocalyxin and a negative correlation with serum mtDNA (R^2^ = 0.855; *p* < 0.0001). Urinary ICAM-1 had a direct correlation with UACR, KIM-1, NAG, and urinary mtDNA, and a negative correlation with eGFR (R^2^ = 0.783; *p* < 0.0001) (Table 2).

### 3.3. Association of Cerebrovascular Hemodynamic Indices with Biomarkers of Podocyte Damage and PT Dysfunction

In univariable regression analysis, IMT-CCAs, PI-ICAs, and PI-MCAs, as well as RI-ICAs and RI-MCAs correlated directly with UACR, synaptopodin, podocalyxin, NAG, and KIM-1, and negatively with eGFR. The CVR evaluated by the BHI showed a direct correlation of this parameter with eGFR and a negative correlation with UACR, synaptopodin, podocalyxin, KIM-1, and NAG (Table 3). This data was introduced into a multivariable regression analysis together with the results obtained by a univariable regression analysis applied to mtDNA and markers of inflammation. The models provided are presented in Section 3.4 and Section 3.5.

### 3.4. Atherosclerosis and Arteriosclerosis of Cerebral Vessels Are Related to mtDNA Changes and to Proinflammatory Cytokines IL-17A, IL-18, TNFα, and ICAM-1 and the Anti-Inflammatory Cytokine IL-10

In univariable regression analysis, IMT-CCAs, PI-ICAs, PI-MCAs, RI-ICAs, and RI-MCAs showed a direct correlation with urinary mtDNA, serum and urinary IL-17A, serum and urinary IL-18, serum and urinary TNFα, and ICAM-1, while there were negative correlations of these hemodynamic indices with serum mtDNA and serum and urinary IL-10 (Table 4 and Figure 2a–f and Figure 3a–f).

The models resulting from multivariable regression analysis for the cerebral hemodynamic indices and mtDNA, interleukins, TNFα, ICAM-1, and renal parameters revealed direct correlations of IMT-CCAs with serum IL-17A and indirect correlations with serum mtDNA and serum IL-10 (R^2^= 0.706; *p* < 0.0001). PI-ICAs correlated directly with serum IL-17A and negatively with serum IL-10 (R^2^ = 0.567; *p* < 0.0001); PI-MCAs showed a direct correlation with serum IL-17A and an indirect one with serum IL-10 (R^2^ = 0.642; *p* < 0.0001). RI-ICAs showed a direct correlation with UACR, serum and urinary IL-17A, serum and urinary IL-18, NAG, serum and urinary ICAM-1, and urinary TNFα; there were negative correlations of RI-ICAs with serum mtDNA and serum and urinary IL-10 (R^2^ = 0.930; *p* < 0.0001). RI-MCAs correlated directly with UACR, urinary mtDNA, serum IL-17A, serum IL-18, NAG, and urinary TNFα, and negatively with serum mtDNA and serum IL-10 (R^2^ = 0.875; *p* < 0.0001) (Table 5). This data shows the impact of mtDNA changes and the inflammatory profile in serum and urine upon the cerebral vessels investigated within the confines of DKD in patients with type 2 DM.

### 3.5. CVR Is Impaired in Patients with Type 2 DM and Correlates with mtDNA Changes and IL-17A, IL-18, IL-10, TNFα, and ICAM-1 Variations

Data derived from the univariable regression analysis shows direct correlations of CVR with serum mtDNA and serum and urinary IL-10 and indirect correlations with urinary mtDNA, serum and urinary IL-17A, IL-18, TNFα, and ICAM-1 (Table 4).

Multivariable regression analysis yielded a model in which BHI correlated directly with serum mtDNA and serum IL-10 and negatively with serum ICAM-1, IL-17A, and urinary NAG (R^2^ = 0.823; *p* < 0.0001) (Table 5). These results point to a significant association of the impaired CVR with mtDNA and inflammation, in conjunction with a biomarker of PT dysfunction, namely NAG, which is very sensitive in defining early DKD. (Figure 2f and Figure 3f).

## 4. Discussion

In the current study, we documented an association of cerebral vessel remodeling in terms of atherosclerosis, arteriosclerosis, and cerebrovascular reactivity with an underlying mtDNA and inflammatory profile in early DKD in patients with neurologically asymptomatic type 2 DM. Furthermore, the inflammatory background, substantiated by the assessment of IL-17A, IL-18, IL-10, TNFα, and ICAM-1, and the instrumental intervention of mtDNA in early DKD and cerebrovascular remodeling were cornered by the significant correlations of these parameters with biomarkers of podocyte damage and PT dysfunction, as well as with cerebrovascular hemodynamic indices, even in the normoalbuminuric stage of DKD.

### 4.1. Mitochondrial DNA Variations in Blood and Urine Are Linked to Atherosclerosis and Arteriosclerosis of Cerebral Vessels in Patients with Type 2 DM and Early DKD

The atherosclerotic and arteriosclerotic cerebrovascular diseases represent an important cause of morbidity and mortality in patients with type 2 DM [2]. The kidney and the brain share vascular, structural, and functional similarities and have common hemodynamic regimens. Therefore, cerebrovascular complications either parallel renal vascular structural changes or, most likely, cerebrovascular remodeling and its functional consequences precede the occurrence of renal microangiopathic complications.

The latter phenomenon was extensively studied, and we demonstrated previously that endothelial dysfunction in cerebral vessels precedes endothelial dysfunction in the kidney in patients with normoalbuminuric type 2 DM, thus implying distinct endothelial territories within the kidney and the brain. Also, we showed that PT dysfunction is dissociated from endothelial dysfunction and represents an early stage of renal involvement in type 2 DM in patients who are still normoalbuminuric [27].

It has been shown that carotid atherosclerosis [35] and renal vascular lesions are more severe in normoalbuminuric DKD [36]. Moreover, normoalbuminuric DKD is associated with an increase in carotid artery IMT concomitantly with an increased intrarenal arterial resistance index, a phenomenon that translates into a generalized process of arteriosclerosis [37].

It should be underlined that asymptomatic subclinical renal and cerebral atherosclerosis is more prevalent and is more rapidly progressive in patients with DKD, as was demonstrated in the NEFRONA study, observations that were related to traditional as well as non-traditional cardiovascular risk factors [38].

In our study, conducted in patients with neurologically asymptomatic type 2 DM, subclinical carotid artery atherosclerosis was significant, as shown by the increased carotid IMT. Also, arteriosclerotic remodeling of cerebral vessels associated with atherosclerosis was reinforced by the increased PIs and RIs in the ICAs and MCAs. Of note, these vascular modifications were detected in all groups of patients studied, including the normoalbuminuric group. The cerebral hemodynamic indices correlated with biomarkers of podocyte injury and PT dysfunction, but most importantly, this correlation was observed in normoalbuminuric patients, thus underlining the fact that albuminuria is not conditional for the development of cerebral and renal vascular modifications. In a prospective study, with a median follow-up of 10.8 years, IMT in the internal carotid artery predicted microalbuminuria development and renal function deterioration [39]. In addition, higher UACR within the normal range was independently associated with early carotid atherosclerosis in patients with type 2 DM, hence low-grade albuminuria contributed to the risk of carotid atherosclerosis [26].

The association of subclinical atherosclerosis investigated by carotid IMT with urinary NAG was signaled in a meaningful study by Kim et al. in patients with type 2 DM, which showed that increased carotid IMT and the presence of carotid plaques correlate positively with urinary NAG. The authors conclude that NAG performed better than albuminuria in the early diagnosis of atherosclerosis [40], which is in agreement with our findings in patients with type 2 DM and early DKD.

Mitochondrial DNA changes associated with inflammatory mechanisms involved in the pathogenesis of DKD may be detected early in the course of DKD, before the occurrence of albuminuria [41]. In a previous study carried out by us in patients with type 2 DM, we found decreased levels of serum mtDNA and increased levels of urinary supernatant mtDNA [13], which is in keeping with the results obtained by Czajka et al. [42] and Jiang et al. [43] in serum, and Wei et al. [12] in the urine. Also, we showed that there was a significant negative correlation of serum mtDNA with podocyte biomarkers synaptopodin and podocalyxin, and with PT dysfunction biomarkers NAG and KIM-1, while urinary mtDNA had a direct correlation with these biomarkers [13].

Mitochondrial dysfunction intervenes in the pathogenesis of atherosclerosis. MtDNA damage modulates the process of atherogenesis from its early subclinical stages [14,19].

In our study, serum mtDNA correlated indirectly with IMT-CCAs, PI-ICAs, PI-MCAs, RI-ICAs, and RI-MCAs, thus showing that low levels of serum mtDNA have a negative impact on cerebral vessels by possibly mediating subclinical atherosclerosis and arteriosclerosis, even in patients with normoalbuminuric type 2 DM with no history or symptoms of cerebrovascular disease. In a prospective 17-year follow-up study performed in middle-aged Swedish women, the authors showed that lower levels of mtDNA copy numbers in serum were independently associated with future risk of cardiovascular disease. The authors assume that mtDNA may have a mediating effect on the association between cardiovascular disease and type 2 DM [44]. Furthermore, in another perspective study, serum mtDNA was related independently to incident cardiovascular events [16]. Decreased leukocyte mtDNA content was associated with atherogenesis [45]. Moreover, decreased mtDNA content in peripheral blood mononuclear cells correlated significantly with atherosclerotic plaques [46]. Mitochondrial DNA was detected in the circulation and vascular cells from atherosclerotic plaques in asymptomatic patients [19]. Also, low mtDNA content in peripheral blood leukocytes was associated with an ischemic stroke [17].

Although in our study in univariable regression analysis serum and urinary mtDNA correlated with the above-mentioned hemodynamic indices, the correlations remained significant in multivariable regression analysis for only IMT-CCAs, RI-ICAs, and RI-MCAs.

These observations may allow for the assumption that the CCAs, ICAs, and MCAs, and the cerebral vascular bed covered by these vessels and their deriving downstream branches, are very sensitive to mtDNA variations in both serum and urine in the course of type 2 DM. Moreover, these results point to a potential molecular mechanism in which mtDNA could be instrumental in subtle changes in the cerebral vessel wall in the early stages of DKD. 

### 4.2. Mitochondrial DNA Changes May Trigger Inflammatory Mechanisms in the Kidney and the Cerebral Vessels

Mitochondrial DNA-associated inflammation plays an important role in the process of atherosclerosis [47] and may contribute to the synthesis of pro-inflammatory cytokines, which further amplify atherosclerotic lesions [20]. In our previous study in patients with type 2 DM with normoalbuminuric DKD, we showed that mtDNA levels in serum and urine display a particular profile in relation to inflammation, investigated by IL-17A, IL-18, and IL-10 within the kidney at both podocyte and proximal tubule levels [13]. In addition to these results, the current study reveals a significant association of serum and urinary mtDNA levels with pro-inflammatory biomarkers TNFα and ICAM-1, which intervene in the pathogenesis of both DKD and cerebrovascular disease. 

Interleukin 17-A (IL 17-A) belongs to the IL-17 family produced by activated T helper (Th) lymphocytes Th17, macrophages, neutrophils, dendritic cells, mast cells, and natural killers. This has pleiotropic functions by promoting the expression of other proinflammatory cytokines [48]. IL-17 could play a role in vascular remodeling, as was shown in rats in carotid artery vascular smooth muscle cells [49]. In a human study, in symptomatic patients undergoing endarterectomy, complicated plaques were significantly associated with increased IL-17A expression [50].

In our study, it should be underlined that IL-17 correlated directly significantly with the cerebrovascular indices in patients with neurologically asymptomatic and early DKD, namely in patients with normoalbuminuric type 2 DM. 

Interleukin-18 (IL-18) is a proinflammatory cytokine expressed by tubular epithelial cells that is involved in the initiation and progression of DKD [51], even in patients with high-to-normal levels of albuminuria [52]. In a study conducted on patients with type 2 DM, carotid IMT correlated directly with high levels of plasma IL-18 [53]. Increased expression of IL-18 has been shown in carotid-vulnerable atherosclerotic plaques [20].

In our patients, IL-18 correlated positively significantly with IMT-CCAs, as well as with PI-ICAs, PI-MCAs, RI-ICAs, and RI-MCAs. In the study by Nakamura et al., patients with type 2 DM had elevated levels of serum and urinary IL-18 as compared to controls. In their study, serum IL-18 correlated with albuminuria and a PT dysfunction biomarker, urinary β2-microglobulin, but also with carotid IMT and brachial-ankle pulse wave velocity. The authors conclude that serum IL-18 may serve as a predictor of the progression of DKD and cardiovascular disease [52]. 

Interleukin-10 is an anti-inflammatory cytokine that alleviates inflammation, subsequent mesangial cell proliferation, and interstitial fibrosis [54]. In a study by Zhu et al., low levels of serum IL-10 were significantly associated with cerebral large artery atherosclerosis and cerebral infarction [55]. Variations in serum IL-10 in patients with ICA stenosis could predict the progression of the degree of stenosis, most likely due to an adaptive mechanism to counteract the activity of pro-inflammatory cytokines [23].

In our study, IL-10 levels correlated negatively with IMT-CCAs, PI-ICAs, PI-MCAs, RI-ICAs, and RI-MCAs. Its levels were decreased in all groups studied and followed a descending trend from normo- to micro- and macroalbuminuria, respectively. The decreased levels found in normoalbuminuric patients show that the anti-inflammatory mechanisms are impaired in the early stages of DKD. Kablak et al. demonstrated that IL-10 was independently associated with atherosclerosis extent evaluated by carotid IMT in a two-year follow-up study [56].

Tumor necrosis factor-alpha is a pro-inflammatory cytokine produced by activated native kidney cells and by activated macrocytes-macrophages [51]. The multiple pleiotropic effects of TNFα are exerted upon all structures of the nephron in DKD, such as glomerular mesangial and endothelial cells, podocytes, and tubular epithelial cells [51]. The serum and urinary levels of TNFα are increased from the early stages of DKD and rise progressively from normo- to macroalbuminuria by activating other local inflammatory pathways [57].

In our study, it has been shown that increased levels of TNFα were increased in all patients studied, including the normoalbuminuria group, a fact that is in keeping with the observation by Chen YL. et al., who showed that urinary and renal levels of TNFα precede the increase in albuminuria [57]. The increased levels of serum and urinary TNFα found in our patients were correlated with podocyte damage biomarkers, a fact that could be explained by the close association of TNFα with podocyte function [51]. Also, we observed a strong correlation of TNFα with tubulointerstitial injury in the early stages of DKD, a fact reported by other studies [51,58]. Significant changes in the levels of TNFα may occur as early as stages 1-2 of DKD, before the initiation of renal function decline [59].

Several studies have revealed the intervention of TNFα in the pathogenesis of atherosclerosis. The study by Kablak-Ziembicka et al. provided data that show that TNFα was related to a two-year cardiovascular event risk in patients with extensive atherosclerosis and arteriosclerosis. The results were supported by additional Doppler ultrasound and imaging methods [56]. 

In our study, TNFα correlated with carotid IMT, RI-ICAs, and the RI-MCAs, thus showing its association with atherosclerotic and arteriosclerotic remodeling of the examined vessels. The intervention of TNFα in the initiation and progression of atherosclerotic carotid lesions was also studied in patients with various degrees of carotid stenosis [23].

Intercellular adhesion molecule-1 is a cell surface glycoprotein expressed by endothelial cells and leukocytes which plays an important role in DKD [60] and cardiovascular disease [61]. 

In our research, serum and urinary levels of ICAM-1 were increased and correlated with biomarkers of podocyte damage and PT dysfunction, in all patient groups, thus emphasizing the intervention of this adhesion molecule in early DKD. 

In the study by Karimi et al., serum ICAM-1 level were increased and followed an ascending trend from normo- to macroalbuminuria [62]. Moreover, these observations hold true in the conclusions of the study by Perlman et al., who showed that the transcript levels of ICAM-1 increase in peripheral blood early in the course of DKD, namely stages 1–2 [59].

In the Atherosclerosis Risk in Communities (ARIC) study, serum ICAM-1 correlated with carotid IMT [63]. Similar data were provided by another study which revealed an association of ICAM-1 with carotid artery atherosclerotic remodeling in patients with hypertensive type 2 DM, but not in normotensive diabetic patients [61]. In patients with transient ischemic attack, ICAM-1 was higher in patients with increased IMT in the carotid arteries [31].

In our patients, ICAM-1 correlated with carotid IMT, the PI-ICAs, PI-MCAs, RI-ICAs, and RI-MCAs in univariable regression analysis, a correlation that remained significant for only the RI-ICAs in multivariable regression analysis, even in the normoalbuminuria stage of DKD. The study by Rubio-Guerra et al., however, showed that circulating levels of ICAM-1 correlated significantly with maximal carotid IMT in patients with normoalbuminuria type 2 DM [22]. Our results in patients with normoalbuminuric type 2 DM may be explained by vascular remodeling in a vascular territory that is mostly affected by atherosclerosis and arteriosclerosis in the early stages of DKD, as documented by the increased RIs which explore the distensibility and resistance in the examined vessels.

### 4.3. Cerebrovascular Reactivity Modifications Are Paralleled by mtDNA Changes and Inflammatory Mechanisms in Normoalbuminuric DKD in Patients with Type 2 DM 

Cerebrovascular reactivity is a hemodynamic parameter that represents the increase in cerebral artery blood flow in response to a vasodilatory stimulus. In our study, CVR was studied under hypercapnia conditions as a vasodilatory stimulus and correlated with mtDNA changes, biomarkers of podocyte damage and PT dysfunction, as well as markers of inflammation. In multivariable regression analysis, however, CVR only correlated directly with serum mtDNA and IL-10 and negatively with serum ICAM-1, IL-17A, and NAG. Due to the scarce data in the literature with regard to CVR and its relation to the biomarkers studied by us, we could compare our results with the study by Martinic-Popovic et al., who showed a negative correlation between serum ICAM-1 and BHI evaluated by transcranial Doppler in the MCAs in patients with transient ischemic attack [31]. Our results are most significant as we found a similar correlation between serum ICAM-1 and BHI, but in neurologically asymptomatic patients with type 2 DM across all study groups staged by albuminuria.

The present study has several limitations. First, this is a cross-sectional study that does not allow for conclusions of causality between mtDNA changes and the inflammatory profile studied and the cerebrovascular modifications, but only for associations between the parameters studied. Second, blood CO_2_ levels were not measured during the breath-holding test to ensure the accuracy of the applied method. Third, all patients were on ACEIs/ARBs and statins, a fact that could have introduced a bias in the interpretation of data. 

However, our study has its strengths. To the best of our knowledge, this study is the first approach to the interrelation between the kidney and brain from a molecular perspective, in which mtDNA and biomarkers of inflammation involved in both organs are discussed in parallel in the attempt to characterize glomerular, tubular, and cerebrovascular territories in patients with type 2 DM and early DKD. The biomarkers investigated and the cerebrovascular hemodynamic indices have substantiated the common underlying mechanisms that intervene in both DKD and cerebral vessel remodeling, even in patients with normoalbuminuric type 2 DM. Also, it is worth mentioning that the results provided by our study point to the cerebrovascular involvement in neurologically asymptomatic patients with type 2 DM from the early stages of DKD. 

## 5. Conclusions

In this study, we documented an association of mtDNA changes in blood and urine with cerebral vessel atherosclerotic and arteriosclerotic remodeling. Also, we observed a significant correlation between pro-inflammatory markers, such as IL-17A, IL-18, TNFα, and ICAM-1, and the anti-inflammatory cytokine IL-10 with hemodynamic cerebrovascular indices assessed by neurosonologic methods. Moreover, we showed that mtDNA modifications in conjunction with the markers of inflammation were instrumental in two closely interlinked organs, the kidney, and the brain, both affected within the confines of macro- and microangiopathic complications of type 2 DM. Of note, this parallel activity of the parameters studied within the kidney and the cerebral vessels occurred in neurologically asymptomatic patients with type 2 DM in the normoalbuminuric stage of DKD. This latter finding draws attention to a particular group of patients with type 2 DM who experience cerebral vessel morphologic and functional modifications and for whom we may assume that the cerebrovascular changes could occur early in the course of DKD.

These observations require longitudinal studies conducted on larger cohorts in order to prove causality between the clinical, renal, and cerebrovascular data. Moreover, in keeping with previous studies on the utility of mtDNA in improving cardiovascular risk stratification in a population-based cohort analysis [16], prospective studies would be useful to validate our findings for further clinical applicability of mtDNA copy numbers in cerebrovascular risk stratification in patients with type 2 DM.

## Figures and Tables

**Figure 1 biomolecules-14-00499-f001:**
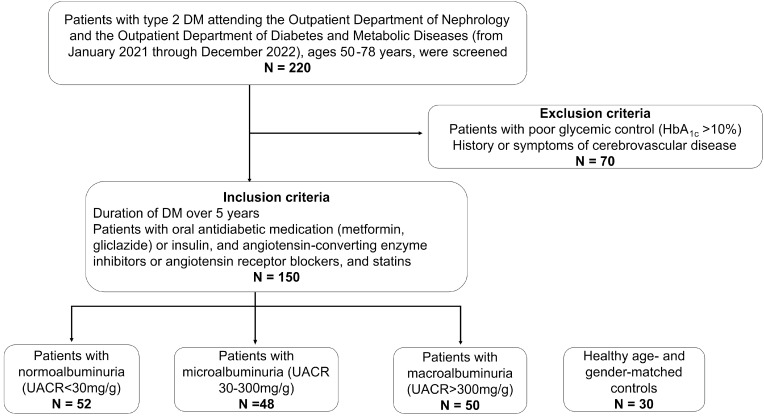
The flow chart that encompasses the study design.

**Figure 2 biomolecules-14-00499-f002:**
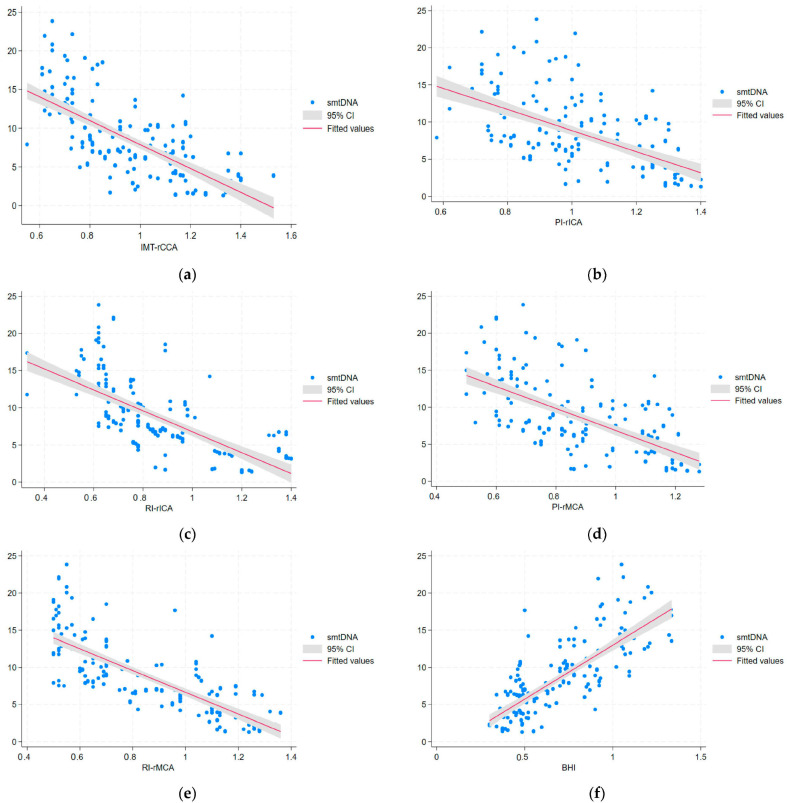
The univariable regression analysis for serum mtDNA (smtDNA) and neurosonological indices such as follows: (**a**) intima-media thickness in the right common carotid artery (IMT-rCCA); (**b**) pulsatility index in the right internal carotid artery (PI-rICA); (**c**) resistivity index in the right internal carotid artery (RI-rICA); (**d**) pulsatility index in the right middle carotid artery (PI-rMCA); (**e**) resistivity index in the right middle carotid artery (RI-rMCA); (**f**) breath holding-index (BHI).

**Figure 3 biomolecules-14-00499-f003:**
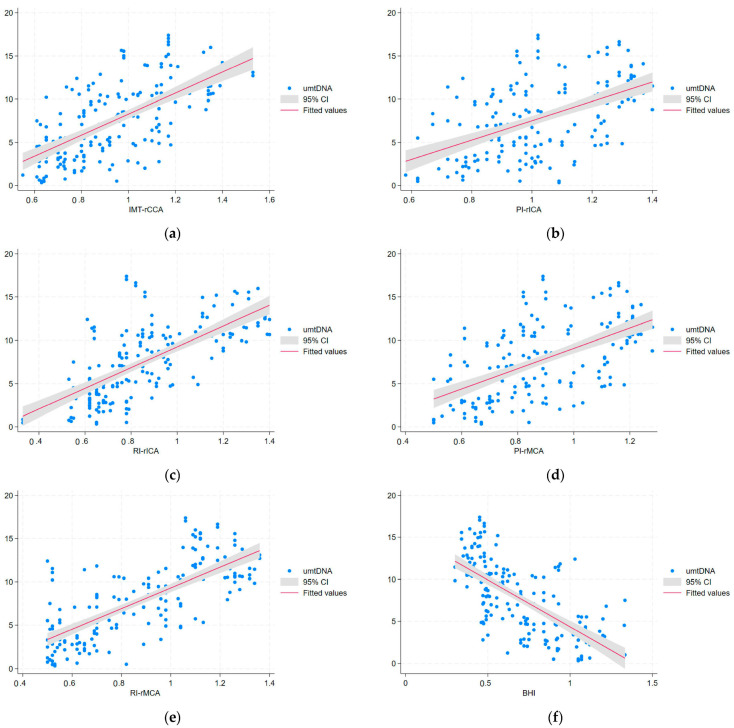
The univariable regression analysis for urinary mtDNA (umtDNA)and neurosonological indices such as follows: (**a**) intima-media thickness in the right common carotid artery (IMT-CCA); (**b**) pulsatility index in the right internal carotid artery (PI-rICA); (**c**) resistivity index in the right internal carotid artery (RI-rICA); (**d**) pulsatility index in the right middle carotid artery (PI-rMCA); (**e**) resistivity index in the right middle carotid artery (RI-rMCA); (**f**) breath holding-index (BHI).

**Table 1 biomolecules-14-00499-t001:** Demographic, clinical, and biological data of patients with type 2 DM and healthy controls.

Parameter	Healthy Controls (N = 30)	Normoalbuminuric Patients (N = 52)	Microalbuminuric Patients (N = 48)	Macroalbuminuric Patients (N = 50)
**Clinical characteristics**				
Age (years)	67.47 (64; 69)	68.33 (65; 72)	69.23 (65; 74)	69.8 (67; 73)
BMI	25.17 (23; 27) ^#,^*	29.26 (26.5; 31.5) ^⌂^	31.47 (28; 34)	30.8 (27; 32)
SBP (mmHg)	117.17 (110; 120) ^#,^*	138.37 (120; 150)	141.42 (130; 152.5)	147.8 (140; 165)
DBP (mmHg)	69 (65; 70) ^#,^*	79.15 (70; 90)	80.2 (70; 90)	81.5 (70; 90)
DM duration (years)	-	15.22 (10; 16.5)	17.73 (12; 23) ^♦^	21.04 (16; 26)
**Biological parameters**				
Hb (g/dL)	13.68 (13.1; 14) *	13.5 (12.75; 14.5) ^♣^	12.4 (11.46; 13.1)	12.39 (11.4; 13.4)
Cystatin C (mg/L)	0.9 (0.8; 1) *	0.95 (0.82; 1.1)	1 (0.88; 1.13) ^■^	1.25 (1.12; 1.38)
Serum creatinine (mg/dL)	0.8 (0.74; 0.85) ^#,^*	0.93 (0.86; 1) ^▲^	0.98 (0.9; 1.02) ^■^	1.33 (1.15; 1.5)
eGFR (mL/min/1.73 m^2^)	84.12 (80.64; 87.85) ^#,^*	76.2 (71.52; 81.21) ^♣^	70.53 (66.88; 73.94) ^■^	50.42 (42.83; 56.56)
HbA1c (%)	5.01 (4.9; 5.1) ^#,^*	7.21 (6.5; 7.65) ^♣^	8.38 (7.2; 9.6)	8.45 (7.8; 9)
Triglycerides (mg/dL)	108.43 (88; 102) ^#,^*	139.95 (102; 171.5)	182.21 (115.5; 215) ^●^	231.4 (160; 296)
Cholesterol (mg/dL)	135.3 (115; 150) ^¶,^*	165.8 (134; 187)	169.4 (134; 211) ^‡^	200.68 (154; 230)
UACR (mg/g)	14.67 (10.19; 17.25) ^#,^*	21.03 (15.10; 27.47) ^♣^	104.5 (60; 139.04) ^■^	1044.16 (464.89; 1365.47)
smtDNA	15.71 (12.87; 17.78) ^#,^*	10.73 (8.21; 12.38) ^♣^	6.92 (5.35; 7.2) ^■^	3.63 (1.82; 4.52)
umtDNA	3.12 (1.08; 4.56) ^ↂ,^*	5.16 (2.79; 7.07) ^♣^	8.27 (6.57; 10.26) ^■^	12.59 (10.69; 14.1)
**Markers of inflammation**				
uIL-18/creat (pg/g)	31.94 (18.27;48.97) ^∆,^*	45.53 (28.06; 60.44) ^♣^	89.44 (69.11; 109.1) ^■^	131.5 (94.18; 162.38)
sIL-18 (pg/mL)	83.5 (64.46; 102.17) ^#,^*	119.78 (94.17; 145.1) ^♣^	153.27 (120.42; 186.46) ^■^	222.03 (154.37; 261.93)
uIL-17A/creat (pg/g)	2.68 (1.94; 2.95) ^#,^*	7.64 (6.18; 9.01) ^♣^	13.89 (11.48; 16) ^■^	27.32 (22; 31.32)
sIL-17A (pg/mL)	4.89 (3.46; 5.54) ^#,^*	10.83 (9.17; 12.3) ^♣^	18.81 (17.48; 20.61) ^■^	37.8 (32.43; 42.3)
uIL-10/creat (pg/g)	15.1 (12.89; 16.38) ^#,^*	8.97 (6.98; 11.23) ^♣^	5.85 (4.72; 6.27) ^⁑^	4.86 (3.71; 5.81)
sIL-10 (pg/mL)	19.75 (17.45; 20.94) ^#,^*	14.07 (12.2; 16.11) ^♣^	11.45 (10.64; 11.98) ^■^	9.54 (8.06; 11.28)
uTNFα/creat (pg/g)	14 (10; 17) ^#,^*	25 (17; 29) ^♣^	39 (29; 50) ^■^	73 (55; 82)
sTNFα (pg/mL)	18 (11; 23) ^#,^*	30 (21; 40) ^♣^	55 (43; 64) ^■^	125 (98; 135)
uICAM-1/creat (ng/g)	2 (1; 2) ^#,^*	3 (3; 4) ^♣^	6 (4; 8) ^■^	13 (8; 15)
sICAM-1 (ng/mL)	2 (1; 2) ^#,^*	5 (4; 5) ^♣^	8 (7; 9) ^■^	16 (12; 19)
**Markers of Podocyte Damage**				
Podocalyxin/creat (mg/g)	38.7 (30.98; 49.55) ^#,^*	65.3 (58.43; 70.72) ^♣^	128.62 (114.42; 152.58) ^■^	520.98 (393.3; 620.54)
Synaptopodin/cr (mg/g)	10.1 (7.44; 11.21) ^#,^*	18.18 (15.38; 21.46) ^♣^	26.55 (24.74; 28.11) ^■^	79.33 (34.75; 133.56)
**Markers of Proximal Tubule Dysfunction**				
NAG/creat (ng/g)	2.02 (1.65; 2.22) ^#,^*	4.74 (2.23; 5.96) ^♣^	12.66 (9.87; 16.09) ^■^	18.52 (16.38; 18.83)
KIM-1/creat (pg/g)	39.3 (27.7; 46.7) ^#,^*	78.37 (66.8; 93.6) ^♣^	134.02 (127.93; 147.68) ^■^	668.5 (595.32; 815.9)
**Cerebrovascular hemodynamic indices**				
IMT-rCCA	0.67 (0.61; 0.73) ^#,^*	0.83 (0.55; 1.07) ^♣^	0.99 (0.76; 1.2) ^■^	1.22 (0.88; 1.53)
PI-rICA	0.82 (0.62; 1.09) ^◊,^*	0.92 (0 58; 1.21) ^♣^	1.07 (0.81; 1.3) ^■^	1.29 (0.95; 1.4)
PI-rMCA	0.60 (0.5; 0.73) ^#,^*	0.81 (0.53; 1.1) ^♣^	0.96 (0.69; 1.19) ^■^	1.17 (0.82; 1.28)
RI-rICA	0.62 (0.33; 0.68) ^#,^*	0.72 (0.61; 0.96) ^♣^	0.89 (0.72; 1.17) ^■^	1.24 (0.78; 1.40)
RI-rMCA	0.53 (0.5; 0.62) ^#,^*	0.66 (0.50; 0.98) ^♣^	0.97 (0.77; 1.23) ^■^	1.20 (1.06; 1.36)
BHI %	1.08 (0.92; 1.33) ^#,^*	0.81 (0.47; 1.09) ^♣^	0.52 (0.42; 0.90) ^■^	0.41 (0.3; 0.59)

Clinical and biological data are presented as medians and IQR, as for variables with skewed distribution. Significance between healthy controls and normoalbuminuric group, ^#^ *p* < 0.001; ^¶^ *p* = 0.001; ^∆^ *p* = 0.005; ^ↂ^ *p* = 0.004; ^◊^ *p* = 0.017; significance between normoalbuminuric group and microalbuminuric group, ^⌂^ *p* = 0.008; ^♣^ *p* < 0.001; ^▲^ *p* = 0.012; significance between microalbuminuric group and macroalbuminuric group, ^♦^ *p* = 0.015; ^■^ *p* < 0.001; ^‡^ *p* = 0.012; ^●^ *p* = 0.008; ^⁑^ *p* = 0.001; significance between healthy controls vs. normoalbuminuric group vs. microalbuminuric group vs. macroalbuminuric group; * *p* < 0.001; BMI: body mass index; SBP: systolic blood pressure; DBP: diastolic blood pressure; DM: diabetes mellitus; eGFR: estimated glomerular filtration rate; UACR: urinary albumin/creatinine ratio; KIM-1/creat: urinary kidney injury molecule-1/creatinine ratio; NAG/creat: N-acetyl-β-(D)-glucosaminidase/creatinine ratio; Podocalyxin/creat: Podocalyxin/creatinine ratio; Synaptopodin/creat: Synaptopodin/creatinine ratio; Hb: haemoglobin; HbA1C: glycated haemoglobin; smtDNA: serum mitochondrial DNA; uDNA: urinary mitochondrial DNA; sIL: serum interleukin; uIL: urinary interleukin/creatinine ratio; sTNFα: serum tumor necrosis factor-alpha; uTNFα: urinary tumor necrosis factor-alpha; sICAM-1: serum intercellular adhesion molecule-1; uICAM-1: urinary intercellular adhesion molecule-1; IMT-rCCA: intima-media thickness right common carotid artery; PI-rICA: pulsatility index right internal carotid artery; PI-rMCA: pulsatility index right middle cerebral artery; RI-rICA: resistivity index right internal carotid artery; RI-rMCA: resistivity index right middle cerebral artery; BHI: breath-holding index.

**Table 2 biomolecules-14-00499-t002:** Multivariable regression analysis for TNFα and ICAM-1 with mtDNA, biomarkers of podocyte injury and PT dysfunction.

Parameter	Variables	Coef β	*p*	95% CI	R²
**uICAM-1**	**UACR**	0.00185	<0.0001	0.001 to 0.002	0.783
	**eGFR**	−0.052	<0.0001	−0.092 to −0.1	
	**KIM-1**	0.006	<0.0001	0.004 to 0.0086	
	**NAG**	0.121	<0.0001	0.057 to 0.184	
**sICAM-1**	**UACR**	0.0021	<0.0001	0.0014 to 0.0029	0.855
	**KIM-1**	0.0028	<0.0001	0.0001 to 0.0056	
	**NAG**	0.129	<0.0001	0.064 to 0.194	
	**smtDNA**	−0.269	<0.0001	−0.354 to −0.183	
	**Podocalyxin**	0.0039	<0.0001	0.0001 to 0.007	
	**Synaptopodin**	0.15	<0.0001	0.001 to 0.0287	
**uTNFα**	**UACR**	0.0072	<0.0001	0.0023 to 0.012	0.683
	**KIM-1**	0.033	<0.0001	0.0197 to 0.047	
	**NAG**	0.654	<0.0001	0.249 to 1.059	
	**umtDNA**	1.038	<0.0001	0.334 to 1.741	
**sTNFα**	**UACR**	0.02	<0.0001	0.015 to 0.025	0.893
	**KIM-1**	0.069	<0.0001	0.057 to 0.082	
	**NAG**	0.949	<0.0001	0.545 to 1.352	
	**smtDNA**	−1.008	<0.0001	−1.557 to −0.46	

sICAM-serum intercellular adhesion molecule; uICAM-1-urinary intercellular adhesion molecule; sTNFα-serum tumor necrosis alpha; uTNFα-urinary tumor necrosis alpha; UACR: urinary albumin/creatinine ratio; eGFR: estimated glomerular filtration rate; KIM-1: urinary kidney injury molecule-1; NAG: N-acetyl-β-(D)-glucosaminidase; smtDNA-serum mitochondrial DNA; umtDNA-urinary mitochondrial DNA.

**Table 3 biomolecules-14-00499-t003:** Univariable regression analysis for cerebral hemodynamic indices with albuminuria, eGFR, and biomarkers of podocyte injury and PT dysfunction.

Parameter	Variables	R²	Coef β	*p*
**IMT-rCCA**	**UACR**	0.36	0.0002	<0.001
	**eGFR**	0.54	−0.012	<0.001
	**Synaptopodin**	0.28	0.003	<0.001
	**Podocalyxin**	0.50	0.0007	<0.001
	**KIM-1**	0.46	0.0005	<0.001
	**NAG**	0.44	0.02	<0.001
**PI-rICA**	**UACR**	0.31	0.0001	<0.001
	**eGFR**	0.36	−0.008	<0.001
	**Synaptopodin**	0.23	0.003	<0.001
	**Podocalyxin**	0.38	0.0006	<0.001
	**KIM-1**	0.36	0.0004	<0.001
	**NAG**	0.35	0.016	<0.001
**PI-rMCA**	**UACR**	0.32	0.0002	<0.001
	**eGFR**	0.41	−0.009	<0.001
	**Synaptopodin**	0.25	0.0029	<0.001
	**Podocalyxin**	0.39	0.0006	<0.001
	**KIM-1**	0.38	0.0004	<0.001
	**NAG**	0.42	0.019	<0.001
**RI-rICA**	**UACR**	0.53	0.0003	<0.001
	**eGFR**	0.54	−0.01292	<0.001
	**Synaptopodin**	0.37	0.004084	<0.001
	**Podocalyxin**	0.54	0.00088	<0.001
	**KIM-1**	0.59	0.00069	<0.001
	**NAG**	0.56	0.025	<0.001
**RI-rMCA**	**UACR**	0.37	0.00027	<0.001
	**eGFR**	0.60	−0.014	<0.001
	**Synaptopodin**	0.36	0.004	<0.001
	**Podocalyxin**	0.62	0.001	<0.001
	**KIM-1**	0.58	0.0007	<0.001
	**NAG**	0.64	0.028	<0.001
**BHI**	**UACR**	0.25	−0.0002	<0.001
	**eGFR**	0.53	0.013	<0.001
	**Synaptopodin**	0.27	−0.0037	<0.001
	**Podocalyxin**	0.44	−0.0008	<0.001
	**KIM-1**	0.43	−0.0006	<0.001
	**NAG**	0.55	−0.026	<0.001

UACR: urinary albumin/creatinine ratio; eGFR: estimated glomerular filtration rate; KIM-1: urinary kidney injury molecule-1; NAG: N-acetyl-β-(D)-glucosaminidase; IMT-rCCA: intima-media thickness right common carotid artery; PI-rICA: pulsatility index right internal carotid artery; PI-rMCA: pulsatility index right middle cerebral artery; RI-rICA: resistivity index right internal carotid artery; RI-rMCA: resistivity index right middle cerebral artery; BHI: breath-holding index.

**Table 4 biomolecules-14-00499-t004:** Univariable regression analysis for the cerebral hemodynamic indices with mtDNA and biomarkers of inflammation.

Parameter	Variables	R²	Coef β	*p*
**IMT-rCCA**	**smtDNA**	0.507	−0.033	<0.001
	**umtDNA**	0.42	0.034	<0.001
	**sIL-17**	0.63	0.014	<0.001
	**uIL-17**	0.58	0.014	<0.001
	**sIL-18**	0.37	0.0019	<0.001
	**uIL-18**	0.37	0.0028	<0.001
	**sIL-10**	0.59	−0.045	<0.001
	**uIL-10**	0.53	−0.049	<0.001
	**sTNF** **α**	0.56	0.004	<0.001
	**uTNF** **α**	0.49	0.007	<0.001
	**uICAM-1**	0.53	0.037	<0.001
	**sICAM-1**	0.57	0.033	<0.001
**PI-rICA**	**smtDNA**	0.36	−0.025	<0.001
	**umtDNA**	0.29	0.026	<0.001
	**sIL-17**	0.49	0.011	<0.001
	**uIL-17**	0.46	0.014	<0.001
	**sIL-18**	0.35	0.0017	<0.001
	**uIL-18**	0.36	0.002	<0.001
	**sIL-10**	0.48	−0.037	<0.001
	**uIL-10**	0.43	−0.034	<0.001
	**sTNF** **α**	0.47	0.003	<0.001
	**uTNFα**	0.41	0.005	<0.001
	**uICAM-1**	0.45	0.031	<0.001
	**sICAM-1**	0.44	0.026	<0.001
**PI-rMCA**	**smtDNA**	0.49	−0.035	<0.001
	**umtDNA**	0.48	0.04	<0.001
	**sIL-17**	0.68	0.015	<0.001
	**uIL-17**	0.56	0.019	<0.001
	**sIL-18**	0.64	0.002	<0.001
	**uIL-18**	0.61	0.003	<0.001
	**sIL-10**	0.75	−0.056	<0.001
	**uIL-10**	0.63	−0.049	<0.001
	**sTNF** **α**	0.51	0.003	<0.001
	**uTNFα**	0.44	0.006	<0.001
	**uICAM-1**	0.49	0.034	<0.001
	**sICAM-1**	0.5	0.029	<0.001
**RI-rICA**	**smtDNA**	0.49	−0.035	<0.001
	**umtDNA**	0.48	0.04	<0.001
	**sIL-17**	0.68	0.015	<0.001
	**uIL-17**	0.56	0.019	<0.001
	**sIL-18**	0.64	0.002	<0.001
	**uIL-18**	0.61	0.003	<0.001
	**sIL-10**	0.75	−0.056	<0.001
	**uIL-10**	0.63	−0.049	<0.001
	**sTNF** **α**	0.69	0.005	<0.001
	**uTNFα**	0.52	0.007	<0.001
	**uICAM-1**	0.76	0.047	<0.001
	**sICAM-1**	0.73	0.04	<0.001
**RI-rMCA**	**smtDNA**	0.62	−0.042	<0.001
	**umtDNA**	0.55	0.046	<0.001
	**sIL-17**	0.75	0.018	<0.001
	**uIL-17**	0.67	0.021	<0.001
	**sIL-18**	0.48	0.002	<0.001
	**uIL-18a**	0.53	0.0039	<0.001
	**sIL-10**	0.64	−0.055	<0.001
	**uIL-10**	0.61	−0.05	<0.001
	**sTNF** **α**	0.55	0.005	<0.001
	**uTNFα**	0.65	0.008	<0.001
	**uICAM-1**	0.64	0.047	<0.001
	**sICAM-1**	0.67	0.041	<0.001
**BHI**	**smtDNA**	0.59	0.041	<0.001
	**umtDNA**	0.46	−0.042	<0.001
	**sIL-17**	0.63	−0.016	<0.001
	**uIL-17**	0.57	−0.019	<0.001
	**sIL-18**	0.37	−0.002	<0.001
	**uIL-18**	0.45	−0.0035	<0.001
	**sIL-10**	0.72	0.057	<0.001
	**uIL-10**	0.71	0.056	<0.001
	**sTNFα**	0.55	−0.004	<0.001
	**uTNFα**	0.52	−0.007	<0.001
	**uICAM-1**	0.53	−0.042	<0.001
	**sICAM-1**	0.57	−0.038	<0.001

smtDNA: serum mitochondrial DNA; umtDNA: urinary mitochondrial DNA; sIL: serum interleukin; uIL: urinary interleukin/creatinine ratio; sTNFα: serum tumor necrosis factor-alpha; uTNFα: urinary tumor necrosis factor-alpha; sICAM-1: serum intercellular adhesion molecule-1; uICAM-1: urinary intercellular adhesion molecule-1; IMT-rCCA: intima-media thickness right common carotid artery; PI-rICA: pulsatility index right internal carotid artery; PI-rMCA: pulsatility index right middle cerebral artery; RI-rICA: resistivity index right internal carotid artery; RI-rMCA: resistivity index right middle cerebral artery; BHI: breath-holding index.

**Table 5 biomolecules-14-00499-t005:** Multivariable regression analysis for cerebral hemodynamic indices and mtDNA, interleukins, TNFα, ICAM-1, and renal parameters.

Parameters	Variables	Coef β	*p*	95% CI	R²
**IMT-rCCA**	**sIL-17A**	0.0088	<0.001	0.0067 to 0.01	0.706
	**sIL-10**	−0.023	<0.001	−0.031 to −0.017	
	**smtDNA**	−0.007	0.002	−0.003 to −0.016	
**PI-rICA**	**sIL-17A**	0.0067	<0.001	0.004 to 0.009	0.567
	**sIL-10**	−0.021	<0.001	−0.029 to −0.013	
**PI-rMCA**	**sIL-17A**	0.0088	<0.001	0.0067 to 0.01	0.642
	**sIL-10**	−0.027	<0.001	−0.034 to −0.019	
**RI-rICA**	**UACR**	0.000049	0.002	0.000018 to 0.00007	0.932
	**NAG**	0.0035	0.003	0.001 to 0.005	
	**uICAM-1**	0.0181	<0.001	0.011 to 0.024	
	**sICAM-1**	0.008	0.008	0.013 to 0.002	
	**uTNF** **α**	0.001	0.001	0.002 to 0.0006	
	**smtDNA**	−0.003	<0.001	−0.001 to −0.005	
	**uIL-17A**	0.008	<0.001	0.012 to 0.003	
	**sIL-17A**	0.008	<0.001	0.004 to 0.012	
	**uIL-18**	0.0007	<0.001	0.0004 to 0.001	
	**sIL-18**	0.0004	<0.001	0.0002 to 0.0007	
	**uIL-10**	−0.027	<0.001	−0.0095 to −0.045	
	**sIL-10**	−0.054	<0.001	−0.074 to −0.035	
**RI-rMCA**	**UACR**	0.000051	0.002	0.000010 to 0.00008	0.875
	**NAG**	0.005	0.001	0.002 to 0.008	
	**smtDNA**	−0.01	<0.001	−0.015 to −0.005	
	**umtDNA**	0.006	0.015	0.001 to 0.122	
	**uTNF** **α**	0.001	0.001	0.0006 to 0.002	
	**sIL-17A**	0.005	<0.001	0.002 to 0.007	
	**sIL-18**	0.0003	<0.001	0.00003 to 0.0005	
	**sIL-10**	−0.01	<0.001	−0.016 to −0.004	
**BHI**	**sICAM1**	−0.006	0.001	−0.001 to −0.018	0.823
	**NAG**	−0.004	0.007	−0.008 to −0.001	
	**smtDNA**	0.008	0.002	0.003 to 0.014	
	**sIL-17A**	−0.005	<0.001	−0.007 to −0.002	
	**sIL-10**	0.004	<0.001	0.024 to 0.037	

UACR: urinary albumin/creatinine ratio; NAG: N-acetyl-β-(D)-glucosaminidase; smtDNA: serum mitochondrial DNA; umtDNA: urinary mitochondrial DNA; uTNFα: urinary tumor necrosis factor-alpha; sIL: serum interleukins; uIL: urinary interleukins; sICAM-1: serum intercellular adhesion molecule-1; uICAM-1: urinary intercellular adhesion molecule-1; PI-rICA: pulsatility index right internal carotid artery; PI-rMCA: pulsatility index right middle cerebral artery; RI-rICA: resistivity index right internal carotid artery; RI-rMCA: resistivity index right middle cerebral artery; BHI: breath-holding index.

## Data Availability

The data that support the findings of this study are available from the corresponding author upon reasonable request.

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
