# Peer review of "Mitochondrial DNA and Inflammation Are Associated with Cerebral Vessel Remodeling and Early Diabetic Kidney Disease in Patients with Type 2 Diabetes Mellitus"

_biomolecules, 2024, doi:10.3390/biom14040499_

Round 1

Reviewer 1 Report

Comments and Suggestions for Authors

This manuscript by Ligia et al. investigates a potential association of mitochondrial DNA (mtDNA) and inflammation with cerebral vessels remodeling in type 2 diabetes mellitus (DM) patients. They perform a cross-sectional study involving a cohort of 150 patients divided into three groups (with normo-, micro- and macroalbuminuria) together with age- and gender-matched healthy controls. They perform multivariable and univariable regression analysis to find significant correlations among serum and urine mtDNA levels and other biological parameters, markers of inflammation and of podocyte damage, proximal tubule dysfunction together with cerebrovascular hemodynamic indices. They show an association between cerebral vessels remodeling in terms of atherosclerosis, arteriosclerosis, and cerebrovascular reactivity with underlying mtDNA and inflammatory profiles in type 2 DM patients with early  diabetic kidney disease (i. e. in the normoalbuminuric stage).

The manuscript presents interesting data relevant to the field, which adds to previous published data from the same authors in the same cohort of patients (Petrica, L et al. Int J Mol Sci 662 2023, 24, 9803, doi:10.3390/ijms24129803). Please see my comments below.

Specific comments

- Can the authors please discuss the source of serum/urinary mtDNA? What cells/tissues are releasing free mtDNA to these biological fluids?

- I strongly suggest authors present the correlation data in graphs together with tables, as it may be more appealing visually.

- Primers concentration rather that volume should be stated in the Material and Methods section (line 141).

Author Response

Point-by-point reply

Reviewer 1

This manuscript by Ligia et al. investigates a potential association of mitochondrial DNA (mtDNA) and inflammation with cerebral vessels remodeling in type 2 diabetes mellitus (DM) patients. They perform a cross-sectional study involving a cohort of 150 patients divided into three groups (with normo-, micro- and macroalbuminuria) together with age- and gender-matched healthy controls. They perform multivariable and univariable regression analysis to find significant correlations among serum and urine mtDNA levels and other biological parameters, markers of inflammation and of podocyte damage, proximal tubule dysfunction together with cerebrovascular hemodynamic indices. They show an association between cerebral vessels remodeling in terms of atherosclerosis, arteriosclerosis, and cerebrovascular reactivity with underlying mtDNA and inflammatory profiles in type 2 DM patients with early diabetic kidney disease (i.e. in the normoalbuminuric stage).

The manuscript presents interesting data relevant to the field, which adds to previous published data from the same authors in the same cohort of patients (Petrica, L et al. Int J Mol Sci 662 2023, 24, 9803, doi:10.3390/ijms24129803). Please see my comments below.

Specific comments

- Can the authors please discuss the source of serum/urinary mtDNA? What cells/tissues are releasing free mtDNA to these biological fluids?

Thank you for your pertinent comment.

Damaged mitochondria deriving from tubular and glomerular cells release their content into the extracellular space and subsequently into the systemic circulation. Hence, mtDNA fragments deriving from the systemic circulation are filtered through the glomeruli and actively secreted into the urine. Extracellular mtDNA may be detected and quantified in both serum and urine and may be utilized as a biomarker of mitochondrial dysfunction [ref 12].

- I strongly suggest authors present the correlation data in graphs together with tables, as it may be more appealing visually.

Thank you for your valuable comment. Graphs representing correlations between serum and urinary mtDNA and cerebrovascular indices in univariable regression analysis are presented in Fig 2a-2f, Fig 3a-3f, Section 3.4.

- Primers concentration rather that volume should be stated in the Material and Methods section (line 141).

Thank you for your pertinent observation.

The assessment of mtDNA was performed according to the protocol by Busnelli, et al (ref 32).

Reviewer 2 Report

Comments and Suggestions for Authors

The authors investigate the association of mitochondrial and inflammation markers with vascular remodeling in the context of diabetic kidney disease in T2D. The topic is scientifically relevant and the study and the presentation well structured and presented, with conclusions that conform with the presented data.

The study shows includes a multivariable regresion model for a selected group of variables, whose significance is properly discussed.

Nevertheless, I think that the inclusion of a ROC curve for sensitivity/specificity and the testing of the stratification of the regression model according to disease progression would further improve the interest of the manuscript.

Author Response

Point-by-point reply

Reviewer 2

The authors investigate the association of mitochondrial and inflammation markers with vascular remodeling in the context of diabetic kidney disease in T2D. The topic is scientifically relevant and the study and the presentation well-structured and presented, with conclusions that conform with the presented data.

The study shows includes a multivariable regression model for a selected group of variables, whose significance is properly discussed.

Nevertheless, I think that the inclusion of a ROC curve for sensitivity/specificity and the testing of the stratification of the regression model according to disease progression would further improve the interest of the manuscript.

Thank you for your suggestion. After discussion with our colleague who performed the statistical analysis, we reconsidered your suggestion. The conclusion is that we cannot perform the ROC analysis due to the following reason:

One of the methods used to determine the distinctiveness of the test in the medical decision-making process is the receiver operating characteristic (ROC) curve method. The ROC curves are used when the dependent variable is dichotomous, whereas the independent variable to be used in decision making is continuous. In our analysis, the dependent variable is continuous.

Reviewer 3 Report

Comments and Suggestions for Authors

This manuscript demonstrated the relationship between mtDNA and various inflammatory markers with cerebrovascular hemodynamics and vascular remodeling in patients with T2D. While the data presented in the manuscript are extensive, there is a lack of novelty. My concerns are as follows:

1. The primary concern of this article is the lack of novelty. The manuscript appears overly similar to another paper published by the same laboratory in IJMS in 2023, both discussing the relationship between mtDNA and multiple inflammatory markers in T2D. Although this manuscript adds result about the connection between these markers and cerebrovascular remodeling and reactivity modification in T2D, changes in cerebral vasculature in T2D have been reported by several laboratories [10.7863/jum.2006.25.5.599] [10.2337/diabetes.54.9.2638] [10.1371/journal.pone.0056264]. Studies investigating the relationship between T2D and mitochondria and mtDNA are also not uncommon [10.1152/ajpendo.00314.2018]. Similarly, research on the relationship between mtDNA in the cardiovascular system and cardiovascular diseases such as atherosclerosis exists [10.1001/jamacardio.2017.3683]. Therefore, this manuscript merely extends the laboratory's existing work on known research. Additionally, the article fails to emphasize the potential clinical applications of the findings.

2. The English language of the manuscript still needs polish. Long sentences decrease logical flow and hinder reader comprehension. Additionally, the emphasis on p-values in Result 3.2 makes the text confusing.

3. Patient inclusion criteria and grouping could be showed by a simple flowchart to make it clearer.

4. Section 3.1 of the main text seems to lack subtitle.

5. In table 2, when the parameter is uICAM-1, the result of umtDNA as variables is not shown. Similarly, only in sICAM-1 part the result of PT dysfunction is shown.

Comments on the Quality of English Language

The English language of the manuscript still needs polish. Long sentences decrease logical flow and hinder reader comprehension. Additionally, the emphasis on p-values in Result 3.2 makes the text confusing.

Author Response

Point-by-point reply

Reviewer 3

This manuscript demonstrated the relationship between mtDNA and various inflammatory markers with cerebrovascular hemodynamics and vascular remodeling in patients with T2D. While the data presented in the manuscript are extensive, there is a lack of novelty. My concerns are as follows:

  1. The primary concern of this article is the lack of novelty. The manuscript appears overly similar to another paper published by the same laboratory in IJMS in 2023, both discussing the relationship between mtDNA and multiple inflammatory markers in T2D. Although this manuscript adds result about the connection between these markers and cerebrovascular remodeling and reactivity modification in T2D, changes in cerebral vasculature in T2D have been reported by several laboratories [10.7863/jum.2006.25.5.599] [10.2337/diabetes.54.9.2638] [10.1371/journal.pone.0056264]. Studies investigating the relationship between T2D and mitochondria and mtDNA are also not uncommon [10.1152/ajpendo.00314.2018]. Similarly, research on the relationship between mtDNA in the cardiovascular system and cardiovascular diseases such as atherosclerosis exists [10.1001/jamacardio.2017.3683]. Therefore, this manuscript merely extends the laboratory's existing work on known research. Additionally, the article fails to emphasize the potential clinical applications of the findings.

Thank you for your valuable comments. The recommended references and the appropriate changes have been added in the Introduction section [ref 4,8,11,16,28].

Also, the potential clinical applications of the findings have been introduced in the

Conclusion section [ref 16]

  1. The English language of the manuscript still needs polish. Long sentences decrease logical flow and hinder reader comprehension. Additionally, the emphasis on p-values in Result 3.2 makes the text confusing.

          Thank you for your pertinent comment. The Results section 3.2 has been           reorganized. In univariable regression analysis, only significant p-values have   been introduced.

  1. Patient inclusion criteria and grouping could be showed by a simple flowchart to make it clearer.

          Thank you for your suggestions. The flowchart with regard to the selection of the        patients, inclusion/exclusion criteria and grouping has been added (Fig 1).

  1. Section 3.1 of the main text seems to lack subtitle.

          Thank you for your observation. The subtitle for Section 3.1 has been added.

  1. In table 2, when the parameter is uICAM-1, the result of umtDNA as variables is not shown. Similarly, only in sICAM-1 part the result of PT dysfunction is shown.

          Thank you for your valuable comment.

          In Table 2, in the model for uICAM-1, umtDNA did not result as significant and         therefore was not included in the table.

          In the same model, the results of PT dysfunction are included (see NAG, KIM-1).

Round 2

Reviewer 3 Report

Comments and Suggestions for Authors

Thank authors for the response and revisions. The addition of Fig 2 and Fig 3 has significantly improved the manuscript, making the results more visual. However, there are still areas for improvement:

1. The first concern I mentioned in the first review report is the lack of innovation in the manuscript, which cannot be resolved simply by referencing the articles I mentioned. I suggest that the authors strengthen the discussion of the novelty of your work in the Introduction and Discussion sections. Based on the existing various researches on the relationship between mtDNA and inflammatory factors in T2D, why is your article valuable?

2. The authors should not hide data that "did not result as significant" from the table. All data, whether significant or not, should be displayed if they are reliable. The lack of "significance" of the data in this study does not mean that it lacks scientific significance.

3. There are unnecessary cursor icons entering the figure in the flowchart of Fig 1.

Author Response

Point-by-point reply

Reviewer 3

Thank authors for the response and revisions. The addition of Fig 2 and Fig 3 has significantly improved the manuscript, making the results more visual. However, there are still areas for improvement:

  1. The first concern I mentioned in the first review report is the lack of innovation in the manuscript, which cannot be resolved simply by referencing the articles I mentioned. I suggest that the authors strengthen the discussion of the novelty of your work in the Introduction and Discussion sections. Based on the existing various researches on the relationship between mtDNA and inflammatory factors in T2D, why is your article valuable?

Thank you for your pertinent suggestion.

The following information has been added:

Introduction:

Despite a large body of evidence with regard to the complex relation between mtDNA abnormalities and atherosclerosis, the literature lacks data concerning the intervention of mtDNA in cerebral vessels structural modifications in type 2 DM. To date, no clinical or experimental study has approached cerebrovascular remodeling in relation to mtDNA variations in the absence of history or symptoms of cerebrovascular disease.

To the best of our knowledge this is the first study which aims to evaluate, from a molecular perspective, a potential association of mtDNA changes in blood and urine and of biomarkers of inflammation with cerebral vessels structural and functional modifications in neurologically asymptomatic patients with type 2 DM.

Discussion:

In our study, serum mtDNA correlated indirectly with IMT-CCAs, PI-ICAs, PI-MCAs, RI-ICAs and with RI-MCAs, thus showing that low levels of serum mtDNA have a negative impact on cerebral vessels by possibly mediating subclinical atherosclerosis and arteriosclerosis, even in normoalbuminuric type 2 DM patients, with no history or symptoms of cerebrovascular disease.

These observations may allow for the assumption that the CCAs, ICAs and the MCAs, and the cerebral vascular bed covered by these vessels and their deriving down-stream branches are very sensitive to mtDNA variations in both serum and urine in the course of type 2 DM. Moreover, these results point to a potential molecular mechanism in which mtDNA could be instrumental in subtle changes in the cerebral vessels wall from the early stages of DKD.

  1. The authors should not hide data that "did not result as significant" from the table. All data, whether significant or not, should be displayed if they are reliable. The lack of "significance" of the data in this study does not mean that it lacks scientific significance.

Thank you for your valuable comment.

A parsimonious model (few variables) is usually preferred over a complex model (many variables). One way to obtain a parsimonious model from a model with many independent variables consists of iteratively removal of the independent variable least significantly related to the dependent variable (i.e., the one with the highest p-value in an analysis of variance table) until all of them are significantly associated to the response variable.

  1. There are unnecessary cursor icons entering the figure in the flowchart of Fig 1.

Thank you for your observation.

The cursor icons have been removed from the flowchart.
